# Impact of Different Irrigation Methods on the Main Chemical Characteristics of Typical Mediterranean Fluvisols in Portugal

José Telo da Gama [1,*], António López-Piñeiro [2], Luís Loures [1,3] and José Rato Nunes [1]

1. VALORIZA—Research Centre for Endogenous Resource Valorization, Polytechnic Institute of Portalegre, 7300-555 Portalegre, Portugal; lcloures@ipportalegre.pt (L.L.); ratonunes@ipportalegre.pt (J.R.N.)
2. Departamento de Edafologia, UNEX—Universidad da Extremadura, 06006 Badajoz, Spain; pineiro@unex.es
3. Research Centre for Tourism, Sustainability and Well-being (CinTurs), University of Algarve, 8005-139 Faro, Portugal
* Correspondence: jose.gama@ipportalegre.pt

**Abstract:** The sustainable management of Mediterranean agricultural soils, characterized by salinization and low organic matter content, requires a thorough understanding of their temporal and spatial evolution. The focal point of this investigation encompasses an area of 6769 ha within the Portuguese Mediterranean basin, from which as many as 686 topsoil specimens were acquired during the periods 2001/2002 and 2011/2012 for the purpose of scrutinizing soil organic matter (SOM) content, pH measured in water, and electrical conductivity (EC). The methodology employed both classical and geostatistical techniques, and the terrestrial samples were classified in accordance with the irrigation mechanisms in use (namely, drip and sprinkler systems), subsequently juxtaposed with their counterparts in rainfed systems. Predictive maps were generated using the Ordinary Kriging algorithm for spatial interpolation. The findings demonstrate that irrigated Fluvisols displayed lower SOM content compared to rainfed soils, with sprinkler-irrigated soils experiencing a 16.1% decrease and drip-irrigated soils showing a more pronounced 26.6% decrease. Moreover, drip-irrigated soils contained 12.5% less SOM compared to sprinkler-irrigated soils. The pH levels stabilized at around 6.6 in both rainfed and irrigated soils, with no significant differences observed between the irrigation methods. Furthermore, irrigated Fluvisols exhibited higher EC values compared to rainfed soils, with both sprinkler and drip-irrigated soils showing values that were 35.2% higher. These results underscore the impact of irrigation practices on soil properties, including elevated EC values due to increased soil salt accumulation. The study highlights the necessity of considering specific irrigation systems and associated practices to ensure sustainable soil health and productivity. Adopting management approaches that account for these factors is crucial for preserving optimal soil conditions in Mediterranean agricultural systems.

**Keywords:** soil organic matter; Mediterranean soils; salinization; drip irrigation; sprinkler irrigation; rainfed agriculture; pH; electrical conductivity; sustainable soil management

## 1. Introduction

Fluvisols [1], prevalent in Mediterranean agricultural regions, exhibit distinctive features such as a substantial organic matter content and a well-developed horizon structure [2–4]. These soils are highly fertile and well-suited for a diverse range of crops, including cereals, legumes, vegetables, and fruit trees [5–8]. The chemical and physical properties of Fluvisols can be influenced by irrigation, which in turn affects crop yield and quality [7,8]. Traditionally, surface irrigation methods, such as flood or furrow irrigation, have been employed in the Mediterranean region for irrigating Fluvisols [9–11]. However, there is a growing shift towards more efficient irrigation techniques such as sprinkler and drip irrigation, aimed at conserving water and enhancing crop production [12–14]. Sprinkler and drip irrigation methods offer several advantages, including reduced water loss through evaporation and

improved water use efficiency [15–17]. The application of these methods to Fluvisols has yielded varying effects on soil chemical properties. Some studies have reported positive outcomes, indicating that these irrigation techniques can enhance soil pH, nutrient availability, and increased crop yield and quality [8,18–20]. Hondebrink et al. [19], in a study conducted in Valencia, Spain, on a Fluvisol with pH values ranging from 7.48 to 7.75, electrical conductivity (EC) between 0.172 and 0.297 dS m$^{-1}$, CaCO$_3$ between 21.0% and 50.7%, soil organic matter (SOM) content between 2.03% and 10.6%, and for a duration of 20 years, analyzed the effect of flood irrigation and drip irrigation with good quality water (i.e., pH between 7.27 and 7.57, EC between 0.635 and 0.755 dS m$^{-1}$, and SAR between 0.450 and 1.28) in six organically managed orchards and six conventionally managed orchards. The main conclusions of the study were that the N and SOM content is higher in soils with organic production, especially when irrigated by drip irrigation: conventional production + drip irrigation (total N = 0.180% and soil organic carbon (C) = 4.72%), organic production + drip irrigation (total N = 0.590% and C = 9.15%), conventional production + flood irrigation (total N = 0.170% and C = 7.56%), and organic production + flood irrigation (total N = 0.290% and C = 8.63%).

Other studies, like the one from Borsato et al. [21], have reported that drip irrigation yields 15% less biomass than sprinkler irrigation when comparing the management of three irrigation systems during a maize cropping season in Italy. Despite the existing body of literature, further research is necessary to thoroughly investigate the effects of different irrigation methods on Fluvisol properties in the Mediterranean region. Recent studies have continued to explore the impact of irrigation methods on Fluvisols, encompassing aspects such as soil electrical conductivity, soil conservation, water content, crop yield, and other soil properties [22–25]. In their study, Castanheira et al. [22] demonstrated that sprinkler irrigation may increase soil salinity depending on the water quality used for irrigation and that it may influence maize crop yields in Fluvisols in Portugal. Darouich et al. [23] conducted a combined study in Italy and Portugal with drip irrigation showing that this irrigation method is key for soil conservation and water resources when combined with proper soil management. Durán et al. [24] found that irrigated Fluvisols in Spain presented higher clay content, organic matter content, and EC yet the same pH when compared to rainfed Fluvisols and Salamanca-Fresno et al. [25] compared conservation and conventional sprinkler irrigated maize and cotton in Spain, observing that during the periods of regulated deficit irrigation the use of frequent sprinkler irrigations did not lead to a reduction in soil CO$_2$ emissions, nor did it bring about any changes in abiotic factors in the topsoil for both the control group and the treatment group. Factors such as soil texture, climate, and crop type may also influence the response of Fluvisols to different irrigation methods [2,12]. The complex and multifaceted nature of the impact of irrigation methods on Fluvisols necessitates further research to comprehensively understand the underlying mechanisms. This study aims to contribute to the existing knowledge by examining the influence of sprinkler and drip irrigation on the chemical characteristics of Mediterranean Fluvisols.

## 2. Materials and Methods

### 2.1. Study Area and Sampling

The research was implemented in the Portuguese Caia Irrigation Perimeter in the Mediterranean basin, particularly within the administrative jurisdictions of Elvas and Campo Maior, in proximity to the Portuguese–Spanish frontier (Figure 1).

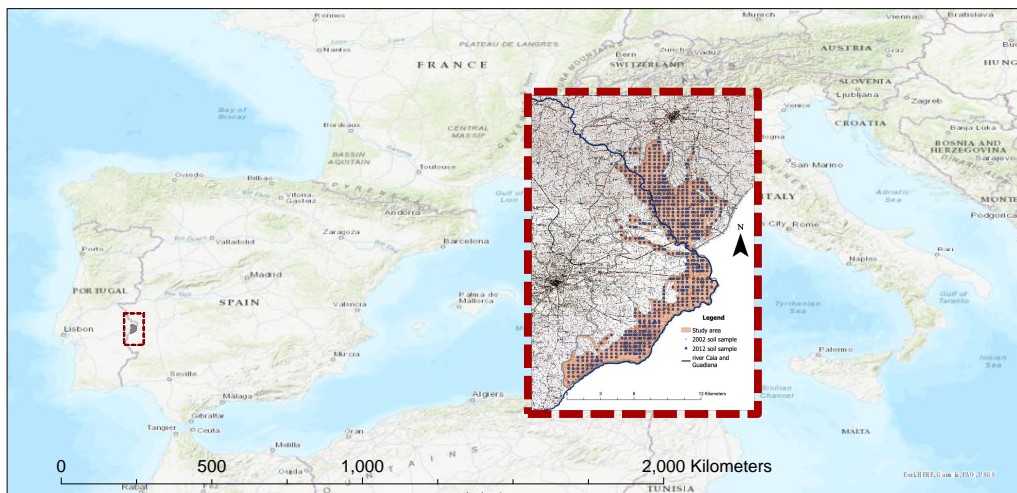

**Figure 1.** Study area in the context of the Mediterranean basin.

The cumulative expanse of the investigative domain encompasses a total of 6769 hectares (Figure 2). The climate in this area is characteristic of the Mediterranean region, with hot dry summers and cool wet winters, classified as Csa according to the Koppen classification. The mean annual precipitation is quantitatively proximate to 483 mm, with the predominant fraction being registered from October through to March, coinciding with the period of lowest temperature fluctuations. July records the highest average monthly temperature at 24.7 °C, while January has the lowest at 8.8 °C [26]. The geology of the study area is heterogeneous, predominantly composed of hyperalkaline and basic rocks [27]. The main crops cultivated in this region include olive trees (*Olea europea* L.), maize (*Zea mays* L.), tomato (*Lycopersicon esculentum* L.), and cereals crops (*Avena sativa* L., *Triticum durum* L., and *Triticosecale cereale* W.). The water quality within the investigative zone has been designated as C1S1 by the Food and Agriculture Organization (FAO), indicative of superior quality characterized by minimal salinity and sodicity [28]. Nevertheless, a marked escalation in bicarbonate ($HCO_3^-$) concentrations has been observed over time. In the 2001/2002 cycle, the average bicarbonate concentration in the study zone was 68.6 mg $L^{-1}$. This figure swelled to 92.0 mg $L^{-1}$ in 2009, and subsequently escalated further to 101.0 mg $L^{-1}$ in the 2011/2012 period. This progressive amplification in bicarbonate concentrations has contributed to an enhanced buffering capacity in subsequent years, representing a shift from low to moderate bicarbonate levels in the irrigation water supply [28]. The classification of the soil in the study area follows the Fluvisol classification according to the World Reference Base of the FAO Soil Resource (WRB) [1].

In 2001, the initial step involved delineating the study area on a Portuguese military chart with a scale of 1:25,000. The boundaries of the study area were drawn to define the specific area to be analyzed: the main Fluvisols in the Caia Irrigation Perimeter. The military map provided the cartographic coordinates, with each square representing 100 hectares. To facilitate the analysis, each main square was divided into nine equal sub-squares, with each sub-square covering an area of 11.11 hectares. The center of each sub-square was georeferenced. In 2001/2002, armed with the coordinates plotted on paper and a GPS system, fieldwork was conducted within the study area. Each of the 655 sites was located, but when a given component of the sub-square in the field appeared heterogeneous with respect to the cropping system (i.e., irrigated or rainfed) or the existing vegetation (e.g., cereals and super-intensive olive groves) that sub-square was divided in half, in the North–South direction, resulting in two new identical polygons of 5.56 ha, with the center of the new polygons being georeferenced. Ten topsoil samples (0–20 cm depth) were collected per site using a stainless steel probe, following a randomized sampling method. In total, 660 samples were collected in 2001/2002 and 686 in 2011/2012. Care was taken to ensure that the samples were thoroughly mixed in situ, resulting in a single composite sample per

site. Subsequently, all samples were air-dried, crushed, sieved to a particle size < 2 mm, and stored for further analysis. At this initial sample collection, the sites were classified as being either irrigated or rainfed [28]. In 2011/2012, building upon the coordinates obtained during the initial sampling period, a new round of sample collection was carried out using the same methodology. At this later stage, the irrigated sites were further categorized as sprinkler or drip irrigation. It is important to note that this study focused exclusively on the sites that maintained a consistent irrigation method throughout the entire study period. Any sites that experienced changes in their irrigation method were excluded from the analysis to ensure uniformity and comparability across the samples. Our methodology involved yearly visits to the sample sites, where we directly observed and recorded the irrigation method in use and the crop being grown. An evolution of the crops in the study fields can be consulted in Figure 3.

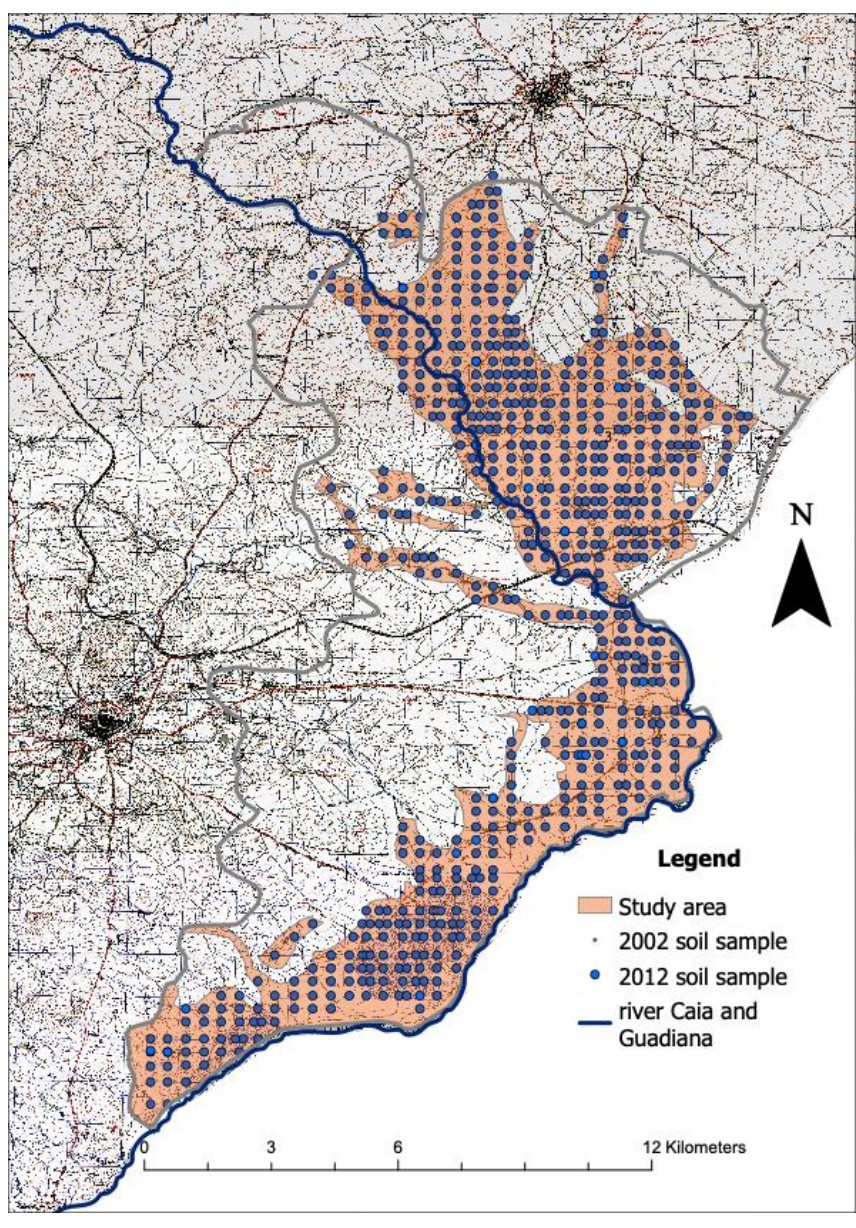

**Figure 2.** Military map with the study area, sampling sites, and rivers Caia and Guadiana.

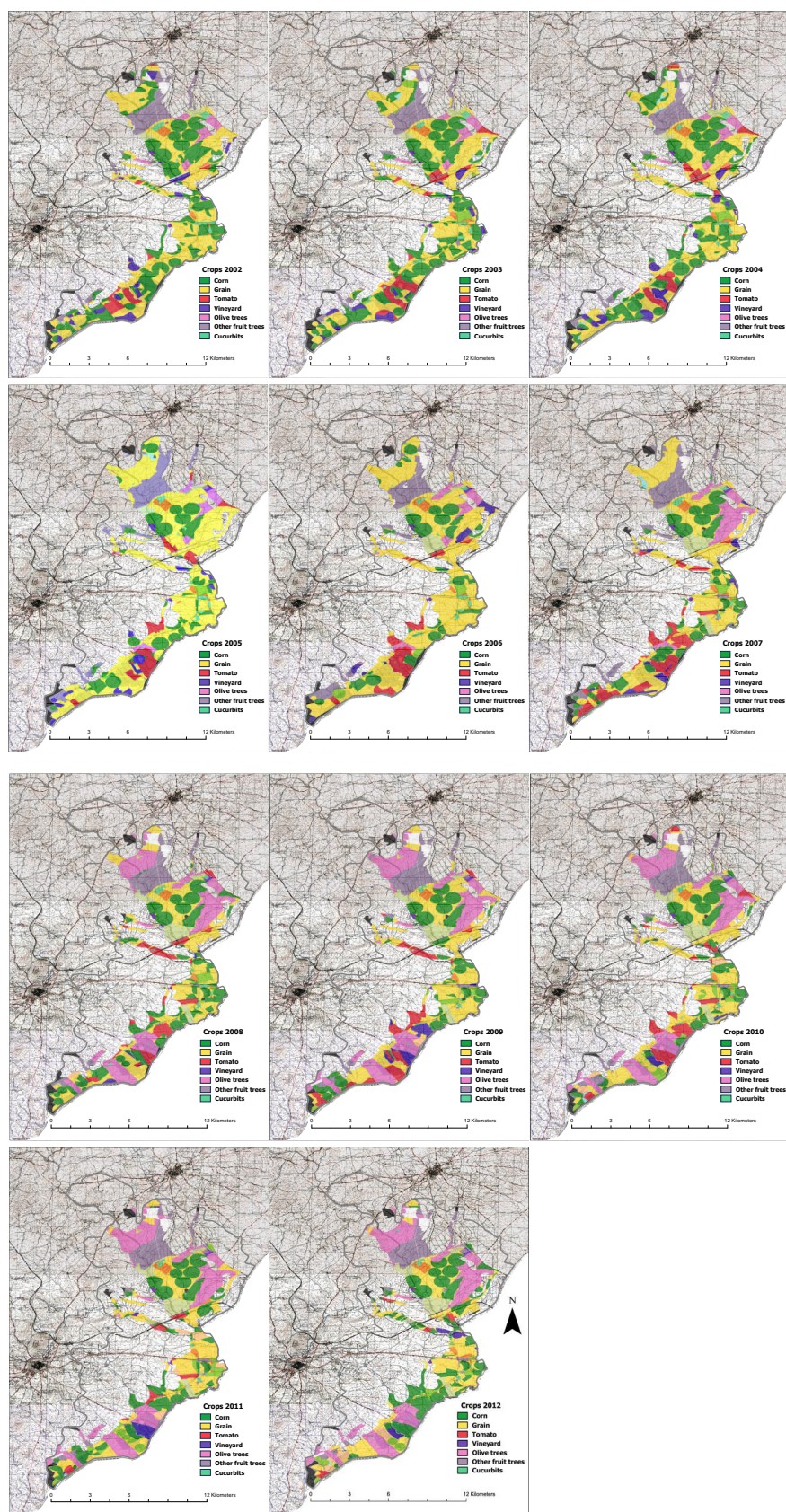

**Figure 3.** Crops present in the study area from 2002 to 2012.

## 2.2. Physico-Chemical Attributes of the Soil and Analytical Methods

In Table 1, we provide the average edaphic properties for the Fluvisols of the study area, encapsulating the physical and chemical characteristics that were analyzed and providing a comprehensive overview of the soil conditions.

**Table 1.** Average edaphic properties of the study area Fluvisols in 2012.

| | Depth | Sand | Silt | Clay | pH | SOM | EC | P | Ca | Mg | K | Na | Ca | Mg | K | Na | CEC | BSP |
|---|---|---|---|---|---|---|---|---|---|---|---|---|---|---|---|---|---|---|
| | (cm) | (%) | (%) | | (water) | (%) | (dS m$^{-1}$) | | | (mg.kg$^{-1}$) | | | | | (cmol$_{(+)}$ kg$^{-1}$) | | | |
| Fluvisols | 0–20 | 72 | 13 | 15 | 6.6 | 1.22 | 156 | 151 | 1640 | 243 | 201 | 45 | 8.3 | 2.1 | 0.40 | 0.17 | 22 | 49 |

SOM: Soil Organic Matter; EC: Electrical Conductivity; CEC: cation exchange capacity (1 M NH4OAc at pH 7.0); BSP: base saturation percentage.

SOM concentrations were obtained through the employment of the wet oxidation technique utilizing potassium dichromate succeeded by titration with ferrous sulfate to gauge the surplus dichromate [29–31]. For the measurement of pH, a blend of soil and water was devised maintaining a volumetric ratio of one part soil to five parts water (1:5 *v/v*). The pH was measured using an MTROHM 692 pH/Ion Meter potentiometer, employing a potentiometric method [31]. Additionally, EC was determined in an aqueous extract, also in a blend of soil and water of one part soil to five parts water (1:5 *v/v*) performed using a WPA CMD 8500 conductivity meter [31,32]. Available calcium and magnesium were assessed using an extraction method involving a solution of ammonium acetate, buffered to pH 7.0, with the addition of a 10% lanthanum chloride solution. The determinations were conducted through atomic absorption spectrophotometry, utilizing flame atomization on a Perkin Elmer Analyzer A300 [33,34]. For the available potassium and sodium, an extraction process was employed using a solution of ammonium lactate and acetic acid, maintained within a pH range of 3.65–3.75 [35]. The subsequent determinations were carried out using atomic absorption spectrophotometry with flame atomization, utilizing the same Perkin Elmer Analyzer A300. The exchangeable cations were extracted utilizing a 1 N NH$_4$OAc (Ammonium acetate) solution, with the pH buffered at 7.0 [36]. Phosphorus content was obtained through extraction using a solution of ammonium lactate and acetic acid, buffered within a pH range of 3.65–3.75 [35]. Subsequent analysis was conducted via molecular absorption spectrophotometry at a wavelength of 650 nm, utilizing a UNICAM UV/VIS UV2 spectrophotometer. The colorimetric analysis was facilitated by the addition of a combined solution of ammonium molybdate and ascorbic acid, which enabled the color development necessary for the measurement.

## 2.3. Statistical and Geostatistical Analyses

The statistical analyses were carried out using the SPSS v.27 software package. To evaluate the data's normality, a comprehensive assessment was conducted, including Shapiro–Wilk tests of normality [37,38], examination of skewness, kurtosis measures, and standard errors [39–41]. These procedures aimed to ascertain whether the data exhibited a normal distribution. Homogeneity of variances was assessed through Levene's tests [42,43], which examined the equality of variances among respective categories. When the data followed a normal distribution with homogeneity of variances, independent sample t-Tests or One-way ANOVA's with Tukey's post hoc results were employed depending on the number of comparisons there were to perform. Furthermore, as the number of samples per subgroup exceeded 30, the central limit theorem was applied to approximate non-normally distributed data to a normal bell curve. Consequently, the aforementioned test was also utilized for non-normally distributed data, provided that the variances were homogeneous. In cases of non-normally distributed data without homogeneity of variances, mean rank (MR) analysis was performed using the Mann–Whitney (U) or Kruskal–Wallis ($X^2$) with pairwise comparisons test depending on the number of comparisons there were to perform. To compare means, a subsample bootstrap with 1000 iterations was conducted, yielding bootstrapped means (BM). All reported results were accompanied by a confidence interval (CI) of at least 95%. Null hypotheses (H0) were rejected when the *p*-value was less than

0.05. Geostatistical analyses were conducted using the ArcGIS PRO 3.1.0 software package. Predictive maps were generated utilizing Ordinary Kriging interpolation, adjusted with a logarithmic factor equation, and supplemented with ancillary variables when available [44].

## 3. Results

### 3.1. SOM

The average soil organic matter (SOM) content, as shown in Table 2, remained stable throughout the study period across the entire study area. Notably, the irrigated Fluvisols exhibited concentrations that were 19.4% lower compared to the rainfed Fluvisols in 2001/2002, consistent with findings from various studies [45–48]. In the Mediterranean basin's edapho-climatic conditions, the increased moisture content in irrigated soils creates an environment conducive to the activity of aerobic organisms. This leads to higher rates of SOM decomposition compared to rainfed soils, and thus, the disparity can be attributed to irrigation favoring conditions for SOM decomposition over synthesis by aerobic organisms [45].

**Table 2.** Temporal evolution of SOM, pH, and across the span from 2001/2002 to 2011/2012.

| Param. | Sampling Date | Irrigation System | Arit. Mean | Sample Size | Statistical Test | *p*-Value |
|---|---|---|---|---|---|---|
| SOM (%) | 2001/2002 | | 1.22 | 660 | T(1344): 0.043 | 0.966 |
| | 2011/2012 | | 1.21 | 686 | | |
| pH (water) | 2001/2002 | all | 6.3 | 660 | T(1344): −3.129 | 0.002 |
| | 2011/2012 | | 6.6 | 686 | | |
| EC (dS m$^{-1}$) | 2001/2002 | | 0.130 | 660 | U: 170,514.000 | 0.000 |
| | 2011/2012 | | 0.156 | 686 | | |

U: Mann-Whitney U test; T: Two-Sample t-Test; all: rainfed plus irrigated cultivation system.

In the most recent sampling (Table 3), the trend persisted, with rainfed Fluvisols displaying higher SOM concentrations compared to the irrigated ones. Specifically, the sprinkler-irrigated soils exhibited concentrations 16.1% lower than the rainfed soils, while the drip-irrigated soils showed a more pronounced difference, with concentrations 26.6% lower. Moreover, a comparison between the two irrigation methods (Table 4) reveals that drip-irrigated Fluvisols contained 12.5% less SOM compared to sprinkler-irrigated ones (results obtained through pairwise comparisons). This is in accordance with the study conducted by Emde et al. [49], who states that the irrigation technique used can significantly influence the impact of irrigation on the content of SOM in farming soils, which is achieved by moderating alterations in the hydrological attributes and physical–chemical properties, and which can differ at various soil depths. Also, considering the highly localized delivery of water and fertilizer in a drip irrigation system, the beneficial impacts on SOM are confined to the regions immediately beneath the drippers [50].

**Table 3.** Evolution of SOM, pH, and EC from rainfed to irrigation system in 2001/2002.

| Param. | Sampling Date | Irrigation System | Arit. Mean | Sample Size | Statistical Test | *p*-Value |
|---|---|---|---|---|---|---|
| SOM (%) | 2001/2002 | rainfed | 1.39 | 266 | U: 35,646.000 | 0.000 |
| | | irrigated | 1.12 | 394 | | |
| pH (water) | 2001/2002 | rainfed | 6.4 | 266 | U: 47,894.500 | 0.060 |
| | | irrigated | 6.2 | 394 | | |
| EC (dS m$^{-1}$) | 2001/2002 | rainfed | 0.102 | 266 | U: 42,338.500 | 0.000 |
| | | irrigated | 0.149 | 394 | | |

U: Mann–Whitney U test.

**Table 4.** Evolution of SOM, pH, and EC from rainfed to sprinkler and drip irrigation system in 2011/2012.

| Param. | Sampling Date | Irrigation System | Arit. Mean | Sample Size | Statistical Test | *p*-Value |
|---|---|---|---|---|---|---|
| SOM (%) | 2011/2012 | rainfed | 1.43 [a] | 249 | $X^2(2)$: 62.755 | 0.000 |
| | | sprinkler | 1.20 [b] | 187 | | |
| | | drip | 1.05 [c] | 243 | | |
| pH (water) | 2011/2012 | rainfed | 6.5 [a] | 249 | $X^2(2)$: 4.169 | 0.124 |
| | | sprinkler | 6.6 [a] | 187 | | |
| | | drip | 6.6 [a] | 243 | | |
| EC (dS m$^{-1}$) | 2011/2012 | rainfed | 0.128 [a] | 249 | $F_{(2676)}$: 6.788 | 0.001 |
| | | sprinkler | 0.173 [b] | 187 | | |
| | | drip | 0.173 [b] | 243 | | |

Superscripts: different superscripts indicate statistical significance within the respective parameter; $X^2$: Kruskal–Wallis test; F: One-way ANOVA.

From 2001/2002 to 2011/2012, the levels of rainfed SOM remained unchanged (Mann–Whitney U: 32,119.000; $N_{2001/2002}$ = 660; $N_{2011/2012}$ = 686; and $p$ = 0.554) at a concentration of circa 1.22%. This observation may indicate the depletion of the active pool of SOM, given the low soil levels observed, with the majority of SOM originating from the passive pool, which undergoes a significantly longer depletion process. The concise interval of sample acquisitions, juxtaposed with the inherently minimal fluctuation of this variable under Mediterranean pedo-climatic circumstances, has facilitated the observed equilibrium of SOM throughout the duration of the investigation [51]. These findings align with the results reported by previous studies [52,53].

*3.2. pH*

During this study, a significant increase in pH was observed in the Fluvisols (Table 2). The pH values showed a 4.8% increase over the study period. However, when comparing the rainfed and irrigated cultivation systems, no significant differences in pH were found, both in 2001/2002 (Table 3) and 2011/2012 (Table 4). In 2011/2012 (Table 4), pH levels stabilized at around 6.6 (results obtained through pairwise comparisons). There are distinct explanations for rainfed and irrigated soils. The stabilization of pH in rainfed soils can be attributed to two main factors. Firstly, the low leaching and drainage capacity of the majority of soils in the study area leads to the saturation of the exchange complex (SEC) with non-acid cations, predominantly calcium ($Ca^{2+}$) and magnesium ($Mg^{2+}$) [54]. This saturation limits the presence of acid cations, such as aluminum ($Al^{3+}$) and hydrogen ($H^+$), in the SEC. Secondly, the intensified climate changes experienced in the Mediterranean basin, characterized by reduced precipitation and elevated temperatures, result in increased evapotranspiration rates compared to rainfall [26]. Consequently, the leaching rates decline, causing fewer non-acid cations to be leached from the soil solution. As a result, these non-acid cations accumulate in the SEC, leading to an elevation in pH [55]. The stabilization of pH in irrigated soils can be explained by similar factors observed in rainfed soils, along with additional considerations. Primarily, the previously delineated influences impacting non-irrigated soils persist in the context of irrigated soils. This is due to the fact that the aggregate of natural rainfall and supplied irrigation water frequently falls short of the requisite volume to facilitate the leaching of non-acid cations from the soil solution [26]. According to the authors consulted [56–58], the accumulation of non-acid cations, along with reduced levels of SOM, is another important form of degradation that occurs in the soils of the Mediterranean basin and that, in semi-arid regions, tends to be accompanied by salinization [59], which, according to Laraus [60], can lead to land abandonment. Consequently, these non-acid cations remain in the soil, allowing them to re-exchange places with the acid cations present in the SEC. Secondly, the irrigation water in the study area experienced a substantial 47% increase in bicarbonate ($HCO_3^-$) content [28]. This accumulation of bicarbonates in the soil solution of irrigated soils leads to the formation of carbonic acid ($H_2CO_3$) through its combination with $H^+$ (Equation (1)). As a result, pH is elevated due to the presence of carbonic acid. The rise in pH is further supported by the study of Iacomino et al. [61].

$$HCO_3^- + H_2O \longrightarrow H_2CO_3 + OH$$
$$H_2CO_3 \longrightarrow H_2O + CO_2$$
(1)

From 2001/2002 to 2011/2012 there was no significant variation in the levels of pH in rainfed soils (Two-sample T(513): $-0.792$; $N_{2001/2002} = 660$; $N_{2011/2012} = 686$; and $p = 0.429$) where pH values remained constant at around 6.5. A previous study by Telo da Gama et al. [28] has reported an upward trend in pH levels within the observed irrigated Fluvisols throughout the study period. This finding provides evidence that long-term irrigation practices have played a role in the observed increase in pH values in these soils. However, no significant correlation was found between soil pH and the specific irrigation methods as also reported by Gul et al. [62] in a study in Pakistan. This observation could potentially be associated with the previously discussed accumulation of non-acid cations within the soil solution, which are not undergoing the process of leaching. The observed pH values align with the expected pH range for Mediterranean soils formed over calcareous substrates [26,63] with the correlation suggesting that the influence of the underlying calcareous material has played a role in shaping the pH characteristics of the studied soils.

### 3.3. EC

During the study period, there was a significant increase in the average electrical conductivity (EC) content within the study area, with values rising by 20.0% (Table 2). Particularly, the irrigated Fluvisols exhibited concentrations that were 46.1% higher than the rainfed Fluvisols in 2001/2002 (Table 3), in accordance with the prevailing consensus within the scientific community [22,64–67]. This increase in EC is also consistent with the observed decrease in SOM levels, as supported by previous studies [54,68,69]. In the most recent sampling (Table 4), the trend of higher EC in irrigated Fluvisols persisted compared to rainfed soils. Both sprinkler and drip-irrigated soils showed values that were 35.2% higher than the rainfed soils, with no significant differences between the two irrigation systems (results obtained via Tukey post hoc test). Over the period 2001/2002 to 2011/2012, the EC levels in rainfed soils remained unchanged, with a concentration of approximately 0.115 dS m$^{-1}$ (Two-sample T(513): $-1.630$; $N_{2001/2002} = 660$; $N_{2011/2012} = 686$; and $p = 0.104$). These results emphasize the strong correlation between EC values and the agricultural system [26,51,54]. Given the constant parameters of irrigation water salinity, soil composition, and local climatic conditions within the study area, the observed correlation may be ascribed to a multitude of factors. These include variations in the cultivated crops (Figure 3) and the corresponding quantities of irrigation water administered to each, the dynamics of water movement wherein the direction and velocity can influence the spatial distribution and accumulation of salts within the soil matrix, insufficient leaching processes potentially exacerbated by elevated evapotranspiration rates, and the specific methodologies employed in the management of irrigation systems, coupled with their respective efficiencies [7,15,23,48]. The restricted leaching potential of Mediterranean terrestrial matrices, in conjunction with the agronomic amplification concomitant with irrigation methodologies—inclusive of fertilizer applications that heighten ionic concentration and consequentially amplify electrical conductivity (EC)—precipitates the phenomenon of secondary salinization in irrigated soil environments [70–72]. The lack of thorough washing of salts by precipitation and irrigation water can be attributed to the aforementioned low leaching capacity of Mediterranean soils and the rising temperatures in the region, exacerbating the disparity between leached and accumulated salts, as demonstrated by Telo da Gama et al. [26].

These findings highlight the profound impact of climate change on soil conditions in the basin, as evidenced by the pH indicator as well, with decreasing precipitation and rising temperatures leading to increased evaporation and reduced leaching rates, resulting in fewer ions being washed away from the soil [15,64,67].

### 3.4. Predictive Maps

The SOM content in the study area exhibits the expected range of low to very low levels, characteristic of the region (Table 5). The prognostic representations (Figure 4) corroborate that the minimum SOM concentrations are scattered across the entire study

area, with particular prominence in regions where there is a geographical overlap with the locations of the Caia River (west) and Guadiana River (east) (Figure 2), which define the majority of the irrigation zones. In contrast, the northern areas, characterized by the predominance of rainfed Fluvisols or regions where olive groves were predominantly planted 5–6 years ago, exhibit the highest SOM content, probably due to the usual practice of incorporating organic matter into the soil when planting olive groves. Notably, there has been a 44% increase in the area covered by Fluvisols with SOM content above 1.50%, rising from 1048 hectares in 2001/2002 to 1500 hectares in 2011/2012 in the whole of the study area. This increase correlates with the expansion of olive groves in the region, as the establishment of new olive groves results in the addition of organic matter to the soil.

**Table 5.** Quantitative measures of SOM (%), pH (water), and EC (dS m$^{-1}$) across two distinct periods—2001/2002 and 2011/2012.

| Parameter | Level | 2001/2002 | | 2011/2012 | |
|---|---|---|---|---|---|
| | | Area (ha) | % | Area (ha) | % |
| SOM (%) | <1.00 | 1264 | 18.7 | 1626 | 24.0 |
| | 1.00–1.25 | 2330 | 34.4 | 2269 | 33.5 |
| | 1.25–1.50 | 2131 | 31.5 | 1373 | 20.3 |
| | 1.50–1.75 | 562 | 8.3 | 778 | 11.5 |
| | >1.75 | 483 | 7.1 | 723 | 10.7 |
| pH (water) | <5.5 | 210 | 3.1 | 364 | 5.4 |
| | 5.5–6.0 | 2406 | 35.5 | 1159 | 17.1 |
| | 6.0–6.5 | 2192 | 32.4 | 1781 | 26.3 |
| | 6.5–7.0 | 1014 | 15.0 | 1673 | 24.7 |
| | 7.0–7.5 | 597 | 8.8 | 950 | 14.0 |
| | 7.5–8.0 | 281 | 4.1 | 455 | 6.7 |
| | >8.0 | 70 | 1.0 | 387 | 5.7 |
| EC (dS m$^{-1}$) | <0.100 | 3526 | 52.1 | 922 | 13.6 |
| | 0.100–0.200 | 2797 | 41.3 | 4438 | 65.6 |
| | 0.200–0.300 | 392 | 5.8 | 1268 | 18.7 |
| | 0.300–0.400 | 53.7 | 0.8 | 135 | 2.0 |
| | >0.400 | 0.0 | 0.0 | 5.9 | 0.1 |

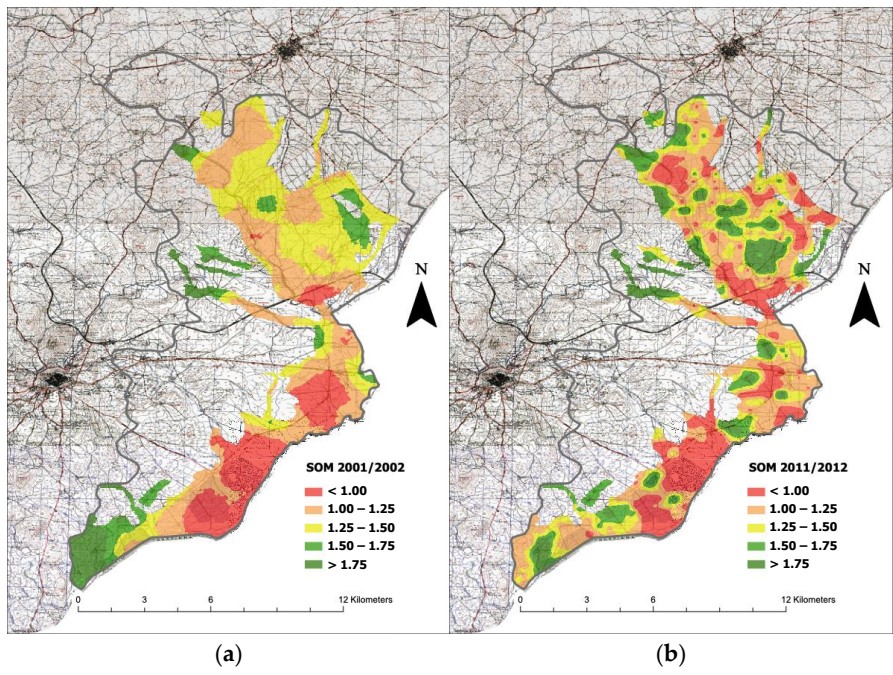

**(a)** **(b)**

**Figure 4.** *Cont.*

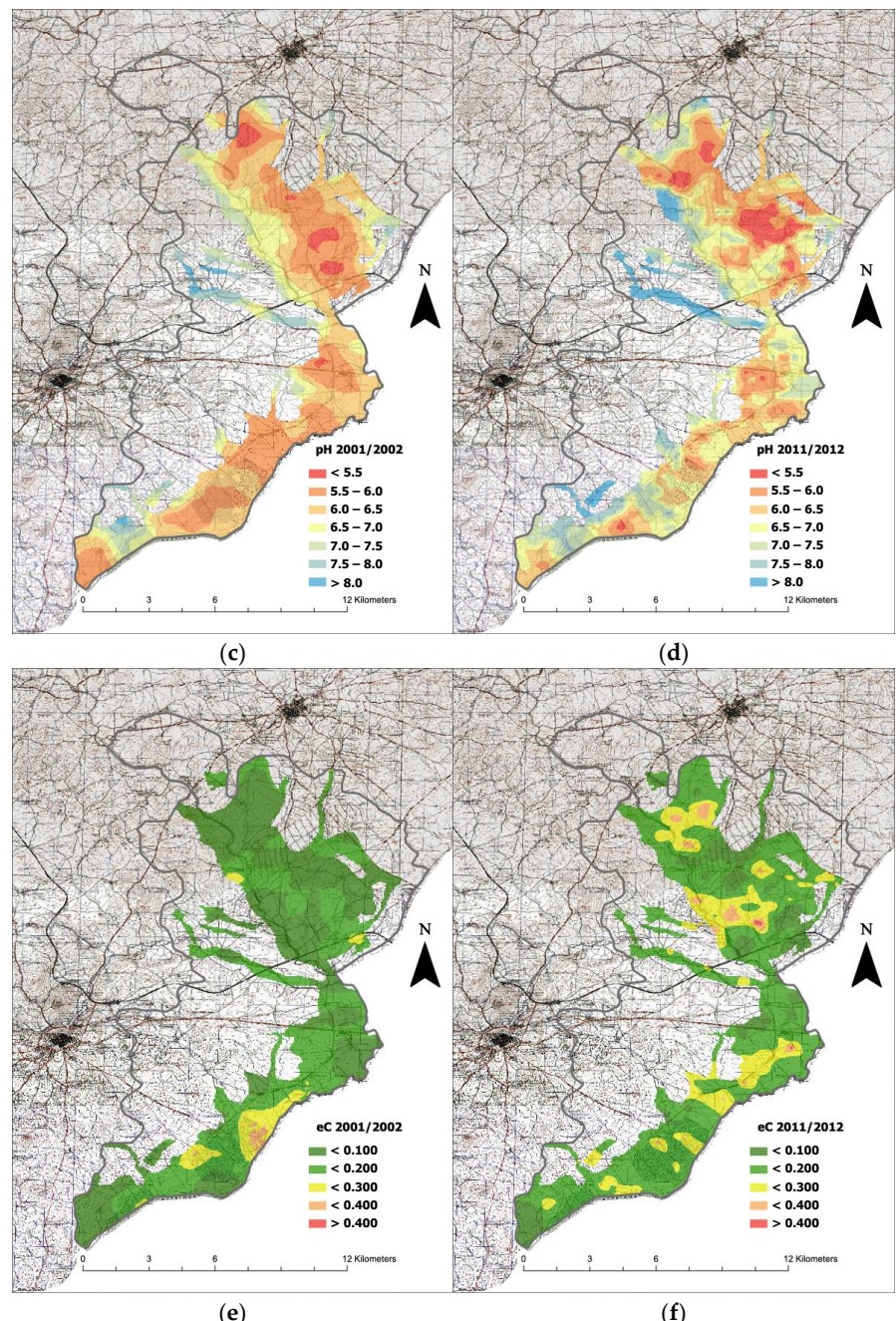

**Figure 4.** Predictive maps for SOM (%) in (**a**) 2001/2002 and (**b**) 2011/2012, pH (water) in (**c**) 2001/2002 and (**d**) 2011/2012, and EC (dS m$^{-1}$) in (**e**) 2001/2002 and (**f**) 2011/2012.

In terms of edaphic pH, a substantial fraction of the investigated soils—specifically, 23.8% in the 2001/2002 period and 38.7% in the 2011/2012 timeframe—demonstrate near neutral values, with pH between 6.5 and 7.5, whilst highly acidic soils, with pH below 5.5, are virtually non-existent. The study region predominantly comprises slightly alkaline soils, with pH above 7.5, which have escalated in their proportional prevalence by 143.1%. The presence of such alkaline soils has implications for crop productivity as it restricts the availability of essential nutrients such as P, Br, Fe, Mn, Zn, Cu, and Co, while simultaneously promoting the availability of potentially harmful elements such as B, Mo, and Se. Soils with pH values ranging from 5.5 to 6.0 decreased by 51.8% during the study period (Figure 4). It is worth noting that the western region of the study area exhibits higher pH values, while the northern, southern, and eastern regions tend to have lower pH values. The

predicted pH increase reported here is consistent with the results previously reported in this paper (Table 2).

Regarding EC, the majority of the soils in both samples show very low conductivity. In 2001/2002, only 6.6% of the study area (445.2 hectares) had EC values exceeding $0.200$ dS m$^{-1}$. Nonetheless, by the cycle of 2011/2012, this proportion had surged to 20.8% (encompassing 1409.0 hectares), marking a substantial escalation of 215.2%. The most elevated electrical conductivity (EC) measurements are discerned within zones characterized by prolonged irrigation practice, whereas the least EC values are localized in regions predominantly governed by rainfed agriculture. Despite the contemporary EC readings within the study domain being considerably distant from categorization as saline (EC > 4.0 dS m$^{-1}$), and hence not of immediate concern [55], they constitute an increasingly significant parameter that could present future complications if the accretion of salts remains unabated.

## 4. Conclusions

The results of this study provide significant insights into the properties of Fluvisols, specifically the soil organic matter (SOM) content, pH, and electrical conductivity (EC), under rainfed conditions and drip and sprinkler irrigation, underscoring the importance of considering the unique effects of different irrigation methods on soil properties. As for SOM content, irrigated Fluvisols, from both sprinkler and drip irrigation methods, exhibited significantly lower concentrations compared to rainfed Fluvisols. The more pronounced difference in drip-irrigated soils suggests that cultivation practices associated with sprinkler irrigation, such as the incorporation of crop residues, may contribute to higher SOM levels. The pH levels stabilized at around 6.6 points, with no significant correlation between pH levels and specific irrigation methods. This indicates that both irrigation methods have similar effects on soil pH, influenced by factors such as saturation of the exchange complex with non-acid cations and bicarbonate accumulation. As for electrical conductivity (EC), the irrigated Fluvisols had significantly higher EC values, highlighting the impact of irrigation practices on soil salts.

Comprehending the differential impacts of various irrigation methods on SOM, pH, and EC can provide critical guidance for farmers in the selection of suitable irrigation techniques. Consequently, these insights hold significant value not only for understanding and managing soil health, but also for implementing sustainable practices and optimizing water usage efficiency. For instance, the observed reduction in SOM in drip-irrigated soils may necessitate the adoption of practices such as the addition of organic matter or the implementation of crop rotation strategies to maintain soil fertility. Furthermore, the disparities in SOM between sprinkler and drip irrigation methods can furnish valuable information for decision-making processes related to water use efficiency.

**Author Contributions:** Conceptualization, J.T.d.G., J.R.N., L.L. and A.L.-P.; methodology, J.T.d.G., J.R.N., L.L. and A.L.-P.; software, J.T.D.G.; validation, J.T.d.G.; formal analysis, J.T.d.G.; investigation, J.T.d.G., J.R.N., L.L. and A.L.-P.; resources, J.T.d.G., J.R.N., L.L. and A.L.-P.; data curation, J.T.d.G.; writing—original draft preparation, J.T.d.G.; writing—review and editing, J.T.d.G., J.R.N., L.L. and A.L.-P.; visualization, J.T.d.G., J.R.N., L.L. and A.L.-P.; supervision, J.T.d.G., J.R.N., L.L. and A.L.-P.; project administration, J.R.N.; funding acquisition, J.R.N. and L.L. All authors have read and agreed to the published version of the manuscript.

**Funding:** The authors acknowledge the financial support of Fundação para a Ciência e a Tecnologia (grant UIDB/05064/2020).

**Data Availability Statement:** The data presented in this study are available on request from the corresponding author. The data are not publicly available due to ongoing research that will be published in subsequent studies.

**Acknowledgments:** This work was supported by national funds through the Fundação para a Ciência e a Tecnologia, I.P. (Portuguese Foundation for Science and Technology) by the project UIDB/05064/2020 (VALORIZA—Research Centre for Endogenous Resource Valorization).

**Conflicts of Interest:** The authors declare no conflict of interest. The funders had no role in the design of the study; in the collection, analyses, or interpretation of data; in the writing of the manuscript, or in the decision to publish the results.

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
