# Peer review of "Impact of Different Irrigation Methods on the Main Chemical Characteristics of Typical Mediterranean Fluvisols in Portugal"

_agronomy, doi:10.3390/agronomy13082097_

Round 1

Reviewer 1 Report

This manuscript describes analysis of three chemical characteristics of fluvisols at a study site in Portugal.  Differences are related to location as well as changes in irrigation practices and cropping patterns over time.  The most recent data were collected more than 10 years ago, so the manuscript describes the historical rather than the contemporary status of the study site.  It is unclear why the authors waited so long to prepare the manuscript, which is of less interest now than would have been the case 10 years ago.  I have a number of suggestions to improve the manuscript (see below).  Each is keyed to line number or to figures and tables.

L1. The title is misleading, because it implies that the scale of the study was larger than was actually the case.  Consider changing it to something like this: Impact of different irrigation methods on the main chemical characteristics of typical Mediterranean fluvisols in Portugal. 

L13. Basin in Portugal

L60. Break up this very lengthy paragraph by starting a new paragraph here.

L87/88. Indicate that the study site is in Portugal (I had to consult a map to determine which side of the “Portuguese-Spanish frontier” you studied). Figure 1 can also be improved.  Consider an inset that locates the study site on a map of Portugal and Spain.  Identify the rivers that are mentioned at L290, and if this is the military map that you refer to in the text, state this in the legend.

L113.  I am curious why this manuscript is being submitted so long after the data were collected (11 years).  This does not change your results, but it means that you are examining changes that occurred in the past and likely do not represent the current status of the study site (2023 maps would almost certainly show additional, more contemporary changes in cropping and irrigation patterns compared to 2012).  Can you justify why you waited so long?

L114. What was your rationale for defining the boundaries of the study area as you did?  Why these boundaries instead of some other boundaries?  There must be a reason.

L116.  Each square—not the study area—was divided into nine sub-squares.

L122. It is unclear what you mean by in situ. Are you telling us that the samples were mixed before you left the field?  Why is in situ significant?

L127/128. It is unclear what you mean by “only the sites that permanently followed the same irrigation system are here presented.”  The original classification was irrigated or non-irrigated, but the sites examined in 2011/2012 were classified as sprinkler or drip—neither of which corresponds to the irrigated designation used earlier.  Please rewrite this sentence to make the basis for selecting the sites presented here very clear.

L129.  This is a small point, but you should state that you visited the study area annually and determined which crops were growing at each site, with the results presented in Figure 2. 

L176/177.  Meaning that irrigation favors aerobic organisms?  If this is your argument, I would state it explicitly.

L183/184. Results of a study indicating…. (who refers to persons, not studies).

Table 1 is quite complicated and confusing.  You use double solid lines to separate the first 6 rows from the second 6 rows from the third 9 rows.  This is clear, but there are inconsistencies in labeling of columns.  For example,  “Sample” for every one of the first six rows is listed, but then for the second six rows, “Sample” is only given for pH.  And for the last 9 rows it is only given for SOM.  It would be much better if you were consistent. Also, the heading for column 2 should be “Sampling Date” rather than “Sample.”  Rather than using sprinkler sometimes and aspersion some other times in the text and table, I would choose one word and use it throughout the manuscript.  The Column 3 heading should be “Irrigation systems” rather than “Cropping systems.”  The third part of the table, which contains 9 rows of data is especially confusing and hard to interpret, because important information is indicated by tiny superscripts that refer back to the legend. So, for example, and if I understand correctly, the first row of SOM data is for areas that were rainfed at both sampling dates, and the second row is for areas that were rainfed at the first sampling date but irrigated by sprinklers at the second sampling date, then rainfed to drip irrigation for the third row.  This is indicated by the little a, b, and c superscripts, which require the reader to refer back to the legend.  But then, when you go down to pH and eC, the superscripts change, although the same samples were analyzed for all three chemical properties.  This means that row 4 is rainfed at both sampling dates, row 5 is sprinkler irrigation at both sampling dates, and row 6 is drip irrigation at both sampling dates.  But this cannot be true, because you tell us that the sprinkler-drip irrigation distinctions were not recorded in 2001/2002.  The last three rows are different again.  Consider separating the last 9 rows into a different table, which could be organized so that the distinctions listed in Column 3 are very, very clear.  You do not want to leave the reader puzzling over what you mean (as a reviewer, I am making a lot of guesses!).  There is also another issue: why is the sample size so variable? At L120, you tell us that there are 655 sites, but here there are sometimes 660 sites, and other times 686 sites.  And other times 679 sites.  You need to explain these differences to the reader.

L208/209. No need to state points.  pH levels simply stabilized at about 6.6.  Also, end this sentence after …comparisons).

L287. Prognostic representations?  Prognostic implies predictions, but aren’t these maps based on data?  Please make this clear.

L290.  See above comment about putting the rivers on the map. 

L292/293.  Is the 44% increase in the area covered by fluvisols with SOM content above 1.5% for the entire study site or the northern areas discussed in the previous sentence? Also, you do not mention the fact that there are lots of red areas in the north where SOM content has decreased a lot over time.  How can this be explained?

L296.  Break up this lengthy paragraph by starting a new paragraph here.

L298. The pH range in parentheses is difficult to interpret.  I think you mean pH between 6.5 and 7.5.  If so, just state it this way.  But according to Table 1, pH values were generally lower than 6.5.  Please make this very clear, so there is no confusion about what you mean.

L307.  Sampled universes?  I think you mean something other than the whole universe, but I do not know what it is.

Table 2.  Spell out parameter in the heading to Column 1 (there is plenty of space), and make the heading for Column 2 Level rather than Interval (all of the values in this column are levels, but only some of them are intervals).

L327/329. Make it: The results of this study provide significant insights into the properties of fluvisols, specifically……, under rainfed conditions and drip or sprinkler irrigation.

L341. Stabilized at about 6.6.

L351/353. This is a very general sentence.  Please elaborate.  Specifically, how are these insights valuable?  And what sustainable practices and specific characteristics do you mean?  It is important that you provide detail here and not just generalities.

The quality of the English is fine--just a few small errors here and there.

Author Response

Thank you for thoroughly review this paper and for raising such important concerns. We have now reviewed the paper as per your suggestions.

L1. The title is misleading, because it implies that the scale of the study was larger than was actually the case.  Consider changing it to something like this: Impact of different irrigation methods on the main chemical characteristics of typical Mediterranean fluvisols in Portugal. 

Thanks. We agree and have updated the title.

L13. Basin in Portugal
updated

L60. Break up this very lengthy paragraph by starting a new paragraph here.
Agree

L87/88. Indicate that the study site is in Portugal (I had to consult a map to determine which side of the “Portuguese-Spanish frontier” you studied). Figure 1 can also be improved. Consider an inset that locates the study site on a map of Portugal and Spain. Identify the rivers that are mentioned at L290, and if this is the military map that you refer to in the text, state this in the legend.
It is now obvious in the text that the location is in Portugal. We have improved Figure 1 and also add another Figure to the paper.

L113.  I am curious why this manuscript is being submitted so long after the data were collected (11 years).  This does not change your results, but it means that you are examining changes that occurred in the past and likely do not represent the current status of the study site (2023 maps would almost certainly show additional, more contemporary changes in cropping and irrigation patterns compared to 2012).  Can you justify why you waited so long?
The dataset collected in both sampled years is extensive and required meticulous and thorough analysis. The 2012 data provide a valuable snapshot of the study site at that time, allowing for a historical comparison with the more recent data that we are collecting at the present moment and that will serve as a critical baseline for understanding long-term trends and patterns.
We believe that the insights derived from the 2012 data remain valuable and contribute to our understanding of the study site's dynamics. Furthermore, we are committed to continuing our research in this area and plan to incorporate more recent data in future studies to provide a comprehensive and up-to-date picture of the changes in cropping and irrigation patterns.

L114. What was your rationale for defining the boundaries of the study area as you did?  Why these boundaries instead of some other boundaries?  There must be a reason.
In order to have rain-fed and irrigated samples we collected the data within the confines of the Caia Irrigation Perimeter.

L116.  Each square—not the study area—was divided into nine sub-squares.
Thanks

L122. It is unclear what you mean by in situ. Are you telling us that the samples were mixed before you left the field?  Why is in situ significant?
We mean that the samples were mixed before we left the field. Mixing the samples in situ, or in their original place, was significant to preserve the integrity and representativeness of each composite sample. By combining the individual samples at the collection site itself, we minimized potential variations and biases that could arise from transporting the samples separately. This approach ensured that each composite sample accurately reflected the characteristics of the specific site from which it was collected.

L127/128. It is unclear what you mean by “only the sites that permanently followed the same irrigation system are here presented.”  The original classification was irrigated or non-irrigated, but the sites examined in 2011/2012 were classified as sprinkler or drip—neither of which corresponds to the irrigated designation used earlier.  Please rewrite this sentence to make the basis for selecting the sites presented here very clear.
It is now clear in the paper.

L129.  This is a small point, but you should state that you visited the study area annually and determined which crops were growing at each site, with the results presented in Figure 2. 
Done.

L176/177.  Meaning that irrigation favors aerobic organisms?  If this is your argument, I would state it explicitly.
We meant that, in the Mediterranean basin's edapho-climatic conditions, the increased moisture content in irrigated soils creates an environment conducive to the activity of aerobic organisms. This leads to higher rates of SOM decomposition compared to rain-fed soils, and thus, the disparity can be attributed to irrigation favoring conditions for SOM decomposition over synthesis by aerobic organisms. This is now clear in the paper.

L183/184. Results of a study indicating…. (who refers to persons, not studies).
It is now clear in the paper.

Table 1 is quite complicated and confusing.  You use double solid lines to separate the first 6 rows from the second 6 rows from the third 9 rows.  This is clear, but there are inconsistencies in labeling of columns.  For example,  “Sample” for every one of the first six rows is listed, but then for the second six rows, “Sample” is only given for pH.  And for the last 9 rows it is only given for SOM.  It would be much better if you were consistent. Also, the heading for column 2 should be “Sampling Date” rather than “Sample.” 
Thanks for pointing this out. As there are no lines separating the collecting date for 2001/2002 or 2011/2012 from the 7th line on we thought  it would be more obvious that they refer to all chemical parameters. We have now modified it.

Rather than using sprinkler sometimes and aspersion some other times in the text and table, I would choose one word and use it throughout the manuscript.  
Thanks. It is now corrected throughout the paper.

The Column 3 heading should be “Irrigation systems” rather than “Cropping systems.
corrected

The third part of the table, which contains 9 rows of data is especially confusing and hard to interpret, because important information is indicated by tiny superscripts that refer back to the legend. So, for example, and if I understand correctly, the first row of SOM data is for areas that were rainfed at both sampling dates, and the second row is for areas that were rainfed at the first sampling date but irrigated by sprinklers at the second sampling date, then rainfed to drip irrigation for the third row.  This is indicated by the little a, b, and c superscripts, which require the reader to refer back to the legend.  But then, when you go down to pH and eC, the superscripts change, although the same samples were analyzed for all three chemical properties.  This means that row 4 is rainfed at both sampling dates, row 5 is sprinkler irrigation at both sampling dates, and row 6 is drip irrigation at both sampling dates.  But this cannot be true, because you tell us that the sprinkler-drip irrigation distinctions were not recorded in 2001/2002.  The last three rows are different again.  Consider separating the last 9 rows into a different table, which could be organized so that the distinctions listed in Column 3 are very, very clear.  You do not want to leave the reader puzzling over what you mean (as a reviewer, I am making a lot of guesses!).  
Thanks on this feedback. We have now created 3 different tables that better state the results and we believe has bring clarity to the issues you presented. As for the superscripts their meaning is that, for the parameter, values are significantly different if the letters are different for each irrigation system. Take SOM for example, rainfed (letter a) is significantly different from sprinkler (letter b) and drip (letter c). Sprinkler (leter b) is significantly different from rain-fed (letter a) and drip (letter c) and drip (letter c) is significantly different from rain-fed (letter a) and sprinkler (letter b). pH is not significantly different for rainfed, sprinkler and drip and, thus, all share the same letter a. Notice that this “letter a” doesn’t have nothing to do with the "letter a” from SOM as this is a different parameter. For eC rainfed (letter a) is significantly different from sprinkler (letter b) and drip (letter b) but no statistically significance was found between sprinkler (letter b) and drip (letter b). We have now subtitled the table to reflect this.

There is also another issue: why is the sample size so variable? At L120, you tell us that there are 655 sites, but here there are sometimes 660 sites, and other times 686 sites.  And other times 679 sites.  You need to explain these differences to the reader.
This is now corrected in the Study area and sampling sub-chapter. Basically, when a given component of the subgrid in the field appeared heterogeneous with respect to the cropping system (i.e., irrigated or rainfed) or the existing vegetation (e.g., cereals and super-intensive olive groves) that subgrid was divided in half, in the North-South direction, resulting in two new identical polygons of 5.56 ha, with the center of the new polygons being georeferenced resulting in more samples taken than the 655 initially planned (660 in 2001/2002 and 686 in 2011/2012).

L208/209. No need to state points.  pH levels simply stabilized at about 6.6.  Also, end this sentence after …comparisons).
corrected.

L287. Prognostic representations?  Prognostic implies predictions, but aren’t these maps based on data?  Please make this clear.
The maps are predictive. They are based on the DB provided (sampled sites) and through krigging algorithms (in GIS) have generated the predictive data between sampled points. This is clearly stated in the Statistical and geostatistical analyses sub-chapter and is common procedure.

L290.  See above comment about putting the rivers on the map. 
Done

L292/293.  Is the 44% increase in the area covered by fluvisols with SOM content above 1.5% for the entire study site or the northern areas discussed in the previous sentence? Also, you do not mention the fact that there are lots of red areas in the north where SOM content has decreased a lot over time.  How can this be explained?
It’s on the totality of the study area. This is now obvious in the text. We re-wrote the paragraph so that it’s more obvious. The influence we believe is due to the places where olive groves were “recently” planted where is usual practice the incorporation of organic matter into the soil, increasing the SOM concentrations in some of the northern areas.

L296.  Break up this lengthy paragraph by starting a new paragraph here.
corrected.

L298. The pH range in parentheses is difficult to interpret.  I think you mean pH between 6.5 and 7.5.  If so, just state it this way.  But according to Table 1, pH values were generally lower than 6.5.  Please make this very clear, so there is no confusion about what you mean.
Exactly. We have corrected the text removing the parentheses. 
Although statistic maps have a statistic category of their own they are consistent with the classical statistic presented in table 1 (they didn’t need to be though as this statistic is performed over the provided data plus the predicted one which may report different results). Here we are predicting that the pH seems to be increasing in area for almost all intervals, which is consistent with the results from lines 3-4 of table 1 (an average of 6.3 in 2001/2002 to 6.6 in 2011/2012). The results reported in Table 1 are but averages, with pH values ranging from 4.X to 8.X. This was now made clear in the paper.

L307.  Sampled universes?  I think you mean something other than the whole universe, but I do not know what it is.
We are referring to “both samples”. Sorry, English is not our main language.

Table 2.  Spell out parameter in the heading to Column 1 (there is plenty of space), and make the heading for Column 2 Level rather than Interval (all of the values in this column are levels, but only some of them are intervals).
Thanks.

L327/329. Make it: The results of this study provide significant insights into the properties of fluvisols, specifically……, under rainfed conditions and drip or sprinkler irrigation.
Done

L341. Stabilized at about 6.6.
done

L351/353. This is a very general sentence.  Please elaborate.  Specifically, how are these insights valuable?  And what sustainable practices and specific characteristics do you mean?  
We’ve changed the results section and this is now clear.

Reviewer 2 Report

Title:    Impact of different irrigation methods on the main chemical characteristics of Mediterranean Fluvisols

Article No:      agronomy-2544413

Journal:            Agronomy

Article Type:   Research Article

Help Documents:        Instructions for Authors for this journal.

Comments to the author (if any):

The review article, titled " The review article, titled " The phytomicrobiome, crop plants and greenhouse gas management.",

by Gama et al., tried to explore the effect of irrigation systems and related agriculture practices on soil health, character, and productivity in Mediterranean agricultural systems. After an the detail review, I highlight the following suggestions and concerns to improve it:

·       After an exhaustive reading of the article, I highlight the following suggestions and concerns to improve it:

·        organic matter content (MOS) check it I think the author means Soil organic matter (SOM

·        This study aims to contribute to existing knowledge by examining the influence of sprinkler and drip irrigation on the chemical characteristics of Mediterranean Fluvisols.

·       It is better to put the locations in quadrants. Study area and sampling

·       This data must be cited, and the meteorological data must be cited

·       The quantification of SOM

·       The correct abbreviation of electrical conductivity is EC, not The eC measurement was performed using

·       The author only writes the heading "Analytical methods “and that’s it. But the physical and chemical traits must be described under the heading of "Physico-chemical attributes of soil" etc and then the detailed protocols which have been used for each feature should be described such as SOM, pH, N, Ca, P, K. ETC.

·        How many times the soil samples were taken to analyze Physico-chemical traits etc

·       Delete it from statistical analysis as well as visual inspections of histograms, normal Q-Q plots, and 150 box plots. These procedures aimed to ascertain whether the data exhibited a normal dis attribution.  

·       The conclusion is highly verbose and complex.  The text should be reduced, and it must be rewritten comprehensively. The prominent results must be highlighted along with the future recommendations  

Author Response

Thanks for reviewing this paper and for your suggestions. We had some difficulties at understanding precisely some of them but we believe we have addressed them all and you will find the paper improved a lot.

organic matter content (MOS) check it I think the author means Soil organic matter (SOM
You are right. Corrected in the paper

·        This study aims to contribute to existing knowledge by examining the influence of sprinkler and drip irrigation on the chemical characteristics of Mediterranean Fluvisols.
·       It is better to put the locations in quadrants. Study area and sampling
An image was added to aid the location of the study area. It was also detailed that the core map provided is a military chart

·       This data must be cited, and the meteorological data must be cited
done

·       The quantification of SOM
corrected

·       The correct abbreviation of electrical conductivity is EC, not The eC measurement was performed using
corrected

·       The author only writes the heading "Analytical methods “and that’s it. But the physical and chemical traits must be described under the heading of "Physico-chemical attributes of soil" etc and then the detailed protocols which have been used for each feature should be described such as SOM, pH, N, Ca, P, K. ETC.
Added in the corresponding section and bibliography updated.

·        How many times the soil samples were taken to analyze Physico-chemical traits etc
This is already described in the materials and methods section. 2 soil samples were taken. One in 2001/2002 and other in 2011/2012.

·       Delete it from statistical analysis as well as visual inspections of histograms, normal Q-Q plots, and 150 box plots. These procedures aimed to ascertain whether the data exhibited a normal dis attribution.  
We need to ascertain if the data is normally or non-normally distributed to select the statistic test to utilize. That’s also stated in the corresponding section. We deleted it though as we already report using Shapiro-Wilk.

·       The conclusion is highly verbose and complex.  The text should be reduced, and it must be rewritten comprehensively. The prominent results must be highlighted along with the future recommendations  
Thanks. It is now corrected in the paper.

Round 2

Reviewer 1 Report

Thank you for responding to all of my original suggestions to improve your manuscript.  Also, I want to compliment you on your English usage.  As a native English speaker, I fully understand the advantages that I enjoy, and I try very hard to provide useful suggestions to scientists who must write in a second language.

OK

Reviewer 2 Report

The authors addressed all the comments and I recommended that the paper can be published in the current issue of Agronomy.